# The Role of the α Cell in the Pathogenesis of Diabetes: A World beyond the Mirror

**DOI:** 10.3390/ijms22179504

**Published:** 2021-09-01

**Authors:** María Sofía Martínez, Alexander Manzano, Luis Carlos Olivar, Manuel Nava, Juan Salazar, Luis D’Marco, Rina Ortiz, Maricarmen Chacín, Marion Guerrero-Wyss, Mayela Cabrera de Bravo, Clímaco Cano, Valmore Bermúdez, Lisse Angarita

**Affiliations:** 1MedStar Health Internal Medicine, Georgetown University Affiliated, Baltimore, MD 21218-2829, USA; msm339@gumed.georgetown.edu; 2Endocrine and Metabolic Diseases Research Center, School of Medicine, Universidad del Zulia, Maracaibo 4002, Venezuela; amanzano_8@hotmail.com (A.M.); olivarlc@hotmail.com (L.C.O.); manuelnava_14@fmed.luz.edu.ve (M.N.); juan.salazar93@gmail.com (J.S.); antioxidante48@gmail.com (C.C.); 3Department of Nephrology, Hospital Clinico Universitario de Valencia, INCLIVA, University of Valencia, 46010 Valencia, Spain; luisgerardodg@hotmail.com; 4Facultad de Medicina, Universidad Católica de Cuenca, Ciudad de Cuenca, Azuay 010105, Ecuador; rortiz@ucacue.edu.ec; 5Facultad de Ciencias de la Salud, Universidad Simón Bolívar, Barranquilla 080022, Colombia; m.chacin@unisimonbolivar.edu.co (M.C.); v.bermudez@unisimonbolivar.edu.co (V.B.); 6Escuela de Nutrición y Dietética, Facultad de Ciencias Para el Cuidado de la Salud, Universidad San Sebastián, Valdivia 5090000, Chile; marion.guerrero@uss.cl; 7City of Houston Health Department, Houston, TX 77054, USA; Mayela.cabrerasebravo@houstontx.gov; 8Escuela de Nutrición y Dietética, Facultad de Medicina, Universidad Andres Bello, Sede Concepción 4260000, Chile

**Keywords:** glucagon, Langerhans’ islets, type 2 diabetes, hyperglycaemia, hypoglycaemia

## Abstract

Type 2 Diabetes Mellitus (T2DM) is one of the most prevalent chronic metabolic disorders, and insulin has been placed at the epicentre of its pathophysiological basis. However, the involvement of impaired alpha (α) cell function has been recognized as playing an essential role in several diseases, since hyperglucagonemia has been evidenced in both Type 1 and T2DM. This phenomenon has been attributed to intra-islet defects, like modifications in pancreatic α cell mass or dysfunction in glucagon’s secretion. Emerging evidence has shown that chronic hyperglycaemia provokes changes in the Langerhans’ islets cytoarchitecture, including α cell hyperplasia, pancreatic beta (β) cell dedifferentiation into glucagon-positive producing cells, and loss of paracrine and endocrine regulation due to β cell mass loss. Other abnormalities like α cell insulin resistance, sensor machinery dysfunction, or paradoxical ATP-sensitive potassium channels (K_ATP_) opening have also been linked to glucagon hypersecretion. Recent clinical trials in phases 1 or 2 have shown new molecules with glucagon-antagonist properties with considerable effectiveness and acceptable safety profiles. Glucagon-like peptide-1 (GLP-1) agonists and Dipeptidyl Peptidase-4 inhibitors (DPP-4 inhibitors) have been shown to decrease glucagon secretion in T2DM, and their possible therapeutic role in T1DM means they are attractive as an insulin-adjuvant therapy.

## 1. Introduction

Type 2 diabetes mellitus (T2DM) is one of the most prevalent chronic and non-communicable diseases worldwide [1]. According to the International Diabetes Federation (IDF), 425 million people were affected by T2DM in 2017, and a fairly conservative projection estimates an overall increase of 693 million cases by 2045 [2]. T2DM is a metabolic disease where insulin resistance (IR), secondary to environmental and genetic factors, has a leading role in its pathophysiology [3,4]. The classic pathophysiological T2DM theorem states that muscle and liver IR is driven by chronic hypoxia, inflammation, and oxidative stress from dysfunctional visceral adipose tissue. In the precursor stages of T2DM, muscle and liver IR stimulates compensatory hyperinsulinemia (CHI) to maintain both fasting and post-prandial glucose levels near the normal range [5].

The last 100 years of research have been guided by an almost-exclusive β-centric vision focused on peripheral insulin sensitivity and β cell stunning and death. Regrettably, the role of the remaining humoral factors was “subordinated and framed” to the IR paradigm. Challenging the dominant IR paradigm, Unger and Orci proposed “the bi-hormonal theory” [6], which suggests the coexistence of relative or absolute hypoinsulinemia with relative hyperglucagonemia in T2DM patients. As mentioned above, clinical, and experimental evidence has shown that T2DM occurs with elevated serum glucagon levels despite hyperglycaemia [7]. Moreover, it has also been observed that peripheral glucagon signalling blockade can normalize serum glucose levels [8], and the use of incretin-mimetics drugs improves glycaemic control via insulinotropic and glucagonostatic effects [9]. Therefore, the main aim of this review is to describe the biology of α-pancreatic cells and their humoral controls to understand the mechanisms of glucagon secretion dysregulation and possible therapeutic targets in hyperglycaemic syndromes and cardiovascular disease management.

## 2. Alpha Cell Physiology: From Secretion to Regulation

The human pancreas contains 1–2 million islets, each measuring 50–100 μm in diameter and containing ∼2000 cells on average. However, islet cells are only 2% of the overall pancreatic mass. It is notable that up to 65% of the human islet cells are α cells [10]. All islet cells originate from the endoderm. Its differentiation into each islet linage is mediated by the *pancreatic and duodenal homeobox 1* (*Pdx1*) and *neurogenin-3* (*Ngn3*) genes. The further evolution of α cells requires both *aristaless-related homeobox* (*Arx*) and *forkhead box protein A2* (*Fox-A2*) action in addition to low expression levels of *paired box 4* (*Pax4*). Other factors important for α cell differentiation include *MAF BZIP Transcription Factor B* (*MafB*), *NK6 Homeobox* (*Nkx6.1*; *Nkx6.2*), *Pax6* [11], and *RNA Paupar* (*PAX6 Upstream Antisense RNA*). The latter is a novel long noncoding that has been shown to regulate α cell development through alternative splicing of *Pax6* [12]. Although the human pancreatic islet has been widely investigated, its cytoarchitecture is still a matter of debate. However, it has been demonstrated that their distribution within islets is not randomized.

In this sense, Orci and Unger in 1975 described the classic model of pancreatic islet architecture for the first time. They proposed that they are grouped in two core cellular clusters [13]: one conformed by β cells located in the central area of islet, and a second group of non-beta cells heterogenous distributed in the periphery [14]. This organization is thoroughly set to allow autocrine, paracrine, and endocrine connections between them to maintain glucose homeostasis [15].

Glucagon is the primary α cell hormone product. It is derived from the proteolysis of the 160-aminoacid pre-pro-glucagon peptide coded by its gene located in 2q24.216. This gene is strongly expressed in α cells, the brain, and L-cells of the gut. Under normal conditions, α cells synthetize glucagon via proconvertase 2 post translational proteolysis, while L-cells produce glucagon-like peptide-1 (GLP-1) via proconvertase 1/3 pathway. Interestingly, some studies demonstrated that alfa cells from diabetic donors can release both GLP-1 and glucagon to enhance insulin secretion [16].

The classic model of glucagon secretion regulation is explained by glucose-medicated glucagon exocytosis. Anatomically, pancreatic islets are highly vascularized to ensure a rapid glucose and aminoacidic sensing. A glycaemic drop near to threshold stimulates glucagon release [17]. The cellular mechanism behind this glucose-dependent regulation of glucagon secretion is described in Figure 1.

Although an intrinsic regulation system clarifies glucose-dependent glucagon secretion, it does not explain the reciprocal regulation between α cells and other pancreatic islet cells or the effect of the autonomic nervous system over glucagon secretion.

## 3. Extrinsic Model for the Regulation of Glucagon Secretion: Neurohormonal Mechanism

It has been demonstrated that certain hormones, ions, and locally secreted neurotransmitters can regulate glucagon release by paracrine, autocrine, or endocrine mechanisms [18,19].

### 3.1. Paracrine Regulation

Omar-Hmeadi et al. observed a lack of inhibition in glucagon exocytosis by hyperglycaemia, somatostatin, or insulin in intact islets in α cells from T2DM cadaveric. Instead, hyperglycaemia inhibits α cell exocytosis, but not in the T2DM donor’s α cell or when paracrine inhibition by insulin or somatostatin is blocked. A reduced Surface expression of Somatostatin-receptor-2 in islet from T2DM donors suggests somatostatin resistance, and consequently, elevated glucagon in T2DM may reflect α cell insensitivity to paracrine inhibition during hyperglycaemia [16].

The critical role of glucagon and acetylcholine (Primary products released by α cell secretory granules) enhancing insulin secretion has already been described [20]. This is possibly a paradoxical fact, given the classical conception that insulin and glucagon are mutually exclusive. Nevertheless, this “conception” is possibly valid in α cells since all β cell products inhibit glucagon secretion during the post-prandial state [20,21]. Furthermore, in vivo studies have shown that insulin signalling activation in α cells can be a critical non-glucose-dependent regulator for glucagon secretion [22,23]. Indeed, a high density of insulin receptors is observed in the α cell membrane [24], and deletion of insulin receptor-coding genes can drive hyperglycaemia and hyperglucagonemia [25]. Although the exact regulatory mechanism in this phenomenon is not fully elucidated yet, a sequential phosphatidylinositol 3-kinase (PI3K) and phosphodiesterase-3B activation with a fall in AMP-c concentration and a final drop in the ATP binding capacity into the K_ATP_ channels have been proposed as a plausible explanatory mechanism [24,26,27]. In addition, potentiation in the G-hydroxybutyric acid (GABA) pathway (described later) by insulin through Akt has also been linked with this process [28,29,30]. Through this, any alteration in this pathway, secondary to oxidative stress or α cell glucotoxicity, could cause the hyperglucagonemia usually seen in T2DM [31].

Other β cells secretory products inhibitory of glucagon secretion are GABA and serotonin via GABA-A and 5-HT1F receptors in α cells [32]. Recent studies suggest that its activation under no-pathologic conditions drives to α cell hyperpolarization, causing VGCA closure and rapamycin target kinase in mammalian cells (mTOR) inactivation [29,30]. Therefore, autoimmune destruction of β cells, a reduction of β cell mass, or impaired secretion could be linked to an increase in α cell mass and hyperglucagonemia in diabetes [30]. On the other hand, serotonin acts by 5-HT1F membrane receptors located in the α cell membrane [33], and their activation generates an intracellular fall in cAMP concentration, inhibiting glucagon release [34]. Almaca et al. [35] proposed that α cells lose their glucose-mediated regulatory property when insufficient serotonergic stimulation is present. Thus, when β cells secretory activity is compromised, α cells lose their “glucose-sensing properties” secondary to scarce paracrine serotonin stimulation, generating a subsequent hyperglucagonemia under hyperglycaemic conditions [35].

Finally, it is essential to mention the role of somatostatin, a classical inhibitory hormone in glucagon secretion [36,37]. Briant et al. [38], in an in vivo study, reported the plausibility of the GAP junction between β cells and δ cells. Thus, β cells glucose-mediated depolarization would subsequently cause coupled-δ cells depolarization, causing somatostatin secretion, α cell hyper-polarization, and inhibition of glucagon release by somatostatin receptor 2 (STTR2) activation. However, Vierra et al. proposed that patients with T2DM may develop two-pore domain K^+^ (K2P) channel TALK-1 disruption on δ cells, and may possibly be an explanation of the altered somatostatin secretion and secondary hyperglucagonemia, as TALK-1 channels are linked with both calcium intake by δ cells and somatostatin exocytosis [39] (Figure 2).

### 3.2. Autocrine Regulation

It is widely known that α cell membranes contain a large number of glucagon receptors (GR) [32]. Furthermore, a recent study reported that GR activation increases glucagon gene transcription via cAMP response element-binding (CREB) activation by the PKA-dependent pathway [40]. Moreover, several studies have reported that blocking GRs has been reported to improve glucose homeostasis [41,42,43], due to glucagon stimulates its own release acting on GR on α cells [44]. In fact, in a study conducted by Liu et al. on αTC1 cells (a pancreatic alpha cell line derived from an adenoma created in transgenic mice), immunofluorescent staining confirmed the presence of GRs on αTC1 cells. After 72 h of treatment with GRs antagonist (to block the effects of the endogenous glucagon), a 44% decrease in αTC1 cell proliferation was observed compared with the control group by counting the cells in the S phase. These results show that glucagon has direct trophic effects on α cells by an autocrine mechanism, and when the pancreatic α-cell number was decreased in db/db mice by a glucagon receptor antagonist, plasma glucagon levels were significantly decreased too [45].

Glutamate is also present in α cell vesicles. It acts as a glucagon-secretion enhancer acting by ionotropic glutamate receptors (iGluR) stimulation or by the binding to other proteins like α-amino-3-hydroxy-5-methyl-4-isoxazolepropionic acid (AMPA) receptor and kainate receptors in the α cell membrane [46]. The activation of iGluR drives to α cell depolarisation, activation of VGCA, and subsequent the fusion of glucagon-containing vesicles with cell membranes [47,48]. Besides, α cells also secrete acetylcholine (ACh), but this neurotransmitter is stored in vesicles different from those containing glucagon (Figure 3). Although ACh effects are not significant in α cell, it is plausible that acetylcholine is co-released with glucagon and improve vascular permeability, promoting glucagon into the circulation since acetylcholine induces nitric oxide (NO) release and thereby causes endothelial-mediated vasodilation mainly through the type 3 muscarinic acetylcholine receptor [49,50].

### 3.3. Juxtacrine Regulation

The juxtacrine mechanism is a common type of signalling between adjacent cells requiring direct contact between cells. Recently, a juxtacrine connection between α- and β-cells has been documented as another exciting way in glucagon secretion control since the confirmation of Eph/ephrin system between α- and β cells [51,52,53]. The Eph signalling system belongs to the superfamily of transmembrane Tyr kinase receptors. Today, it is well established that Eph allows short-distance cell-cell interaction by binding with their specific ligands (ephrin), primarily affecting cytoskeleton and leading to cell repulsion or adhesion in some circumstances. In this regard, the first studies in this field showed that many processes involving fast changes in cellular morphology were ephrin–Eph dependent [54]. More recently, other critical ephrin–Eph signalling-mediated processes have been identified and characterised, e.g., axon guidance, synaptic plasticity, cancer, and processes like juxtacrine hormones release control involving short-distance cell-cell communication. In addition, there is increasing evidence about its influence on cell differentiation, proliferation, and apoptosis regulation [55].

Regarding intra-islet regulation, the association of ephrin ligands from β cells and Eph receptors expressed in α cells seems to be a potent signalling pathway inhibiting glucagon secretion [56]. For instance, Eph receptor A4/7 stimulation on α cells surface by ephrins on β cells correlates with F-actin network maintenance and low glucagon secretion level [51]. Furthermore, EphA receptors knock-out in α cells leads to an altered glucagon release. In addition, isolated α cells display an uncontrolled glucagon release that can be restored by activating the EphA receptor signalling [51,53,56,57].

In the same vein, Hutchens et al. reported a new glucagon secretion model where islet cells inhibit glucagon secretion through α cell tonic stimulation. In this model, ephrin-A ligands on neighbouring islet cells stimulate EphA receptors on α cells to inhibit glucagon secretion. In addition, disruption of EphA4 receptors and EphA forward signalling results in enhanced glucagon secretion and F actin density decrease, while stimulation of EphA forward signalling results in both the inhibition of glucagon secretion and an increase in F actin density. Moreover, sorted α cells without cell-cell contacts display glucagon hypersecretion and lack glucose inhibition of glucagon secretion. In line with this model, restoring EphA forward signalling inhibits glucagon secretion to levels observed in intact islets and restore the inhibition of glucagon secretion by glucose [51].

The EphA4, EphA5, and EphA7 receptors and the ephrin-A5 ligand seem to be essential components of the Eph/ephrin system in regulating glucagon secretion [51,57]. However, the signalling cascade triggered by EphA receptor activation is not entirely understood; it has been found that RhoA, a Rho family GTPase, could be a critical element in the regulation of F-actin networks development in α cells, as it promotes a loss in F-actin density once pharmacologically inhibited [58]. Taking these data as a whole, it is plausible to hypothesise that one of the processes related to the hyperglucagonemia found in diabetes may be linked to this deflective signalling network in the context of a recent finding from the Hughes’ group, who observed a 50% reduction in ephrin-A5 density in human islets from patients with T2DM [51].

Other types of juxtacrine interaction have been suggested to regulate glucagon secretion. One important example is the primary cilium system, a small subcompartment of the cell but has a powerful influence on pancreatic islet function. Since primary cilia located in the endocrine and exocrine cells of the islets have a high signalling potential, researchers have hypothesised that these structures could be involved in the communication between the different islet cell types. Recently, most of the attention has been directed to specific GPCRs localised to primary cilia as a major signalling system in α and β cells. In this regard, loss of cilia disrupts β cell endocrine function, but regrettably, the molecular mechanisms of this effect are still mostly unknown. It has been documented that insulin secretion is impaired when it is disrupted in β cells [59,60,61,62,63,64]. Likewise, it has been proposed that the neural cell adhesion molecule (NCAM) could be another form of cell-cell interaction that would regulate glucagon secretion after NCAM^−/−^ mice, showing a defect in glucagon secretion by an alteration in the exocytosis of its granules, which responds to a lack of reorganisation of the F-actin networks in α cells [65].

### 3.4. Endocrine Regulation

Non-pancreatic hormones also have an active role in α cell regulation. For example, Bagger et al. [66] demonstrated that intravenous (IV) glucose administration did not reduce glucagon levels in subjects with T2DM; however, it was achieved with oral glucose administration, suggesting a potential role of intestine lining and incretins on α cell. In addition, GLP-1 and gastric inhibitory polypeptide (GIP) are hormones encoded by the *GCG* gene (the same encodes glucagon), secreted by enterocytes, with well-studied insulinotropic effects [67]. However, considering the difficulty of these experimental studies, its activity on α cells remains disputed [67,68].

GLP-1 receptors (GLP1R) in α cells have been previously reported, and their activation by GLP-1 drives glucagon secretion inhibition [69,70]. Ramracheya et al. [71], in an in vivo study design, concluded that this inhibitory response occurs by GLP-1-directed actions on α cell via PKA-dependent pathways. They propose that, besides the low density of GLP1R in α cell, their activation could cause a slight reduction in the α cell action potential amplitude by K_ATP_ modulation, producing a significant cell voltage change inhibiting P/Q-type Ca^2+^ channels, and therefore blockade of glucagon exocytosis. This process is independent of glucose levels or insulin and somatostatin stimulation. However, build on the fact that GLP-1 cannot block the glucagon effects during hypoglycaemia, it has been hypothesised that somatostatin mediates the glucagonostatic effect of GLP-1 [38,46].

Recent in vivo studies have reported GLP-1/GLP1R complexes within α cell endosomal vesicles in diabetic mice [46] and a small pulsatile secretion of GLP-1 by α cell [72]. With this framework, it has been suggested that, under pathologic conditions, α cell up-regulates GLP-1 gene expression to be transformed into β cells and provide paracrine GLP-1 stimulation for the remaining β cells to increase their survival as a trophic factor for them [46,73].

The aforementioned results demonstrate that the regulation of glucagon secretion relies on both intrinsic mechanisms in the α cell and extrinsic mechanisms in neighbouring and nearby cells since even autacoids like serotonin exhibit receptors in the α cell membrane inhibiting glucagon release. When α cells have a weak serotonergic tone, they lose their glucose-dependent inhibition of the glucagon-release, generating hyperglucagonemia under hyperglycaemic conditions.

## 4. Hyperglucagonaemia and α Cell Dysfunction in Diabetes: Beyond Insulin and IR

Extensive emerging evidence has been published among the “bi-hormonal theory” in T2DM pathogenesis in which the coexistence of hyperglucagonemia and relative insulin deficiency increase gluconeogenesis and exacerbates peripheral IR [74], leading to overt T2DM development. Currently, there is some consensus regarding hyperglucagonemia origin. This is centred on two possible mechanisms: (1) a progressive loss in the regulatory mechanisms in the secretion patterns due to α cells functional alterations [75,76], or (2) modification in both the islet microarchitecture and cellularity [77,78,79,80].

### 4.1. Structural Alterations in Pancreatic Islet

Conclusive evidence has demonstrated a β cell mass reduction and a concomitant decrease in insulin secretion in subjects with long-standing T2DM [62,63] and a sensible fall in GABA and serotonin release, which are crucial paracrine regulators of glucagon secretion, as explained previously. However, post-mortem studies have reported an increased α cell mass in subjects with diabetes. Nonetheless, the evidence is not conclusive [63]. This finding can be related to a loss of α cell regulating factors or a compensatory mechanism secondary to β cells mass loss [63]. Nevertheless, multiple studies have reported that an elevation of Interleukin 6 (IL-6) circulating levels in T2DM adult mice is possibly linked with an expansion of α cell mass and hyperglucagonemia [64], suggesting that α cell proliferation in Type 1 Diabetes Mellitus (T1DM) is probably IL-6-dependent [81].

Hyperglucagonemia and α cell proliferation have been reported in subjects with T1DM [82,83,84]. Therefore, through those findings, multiple in vivo studies were developed to find pathways that could serve as a therapeutic target in T1DM treatment. In this sense, Zhang et al. [85] reported that the activation of the Sirtuin 1 (SIRT1) pathway, by the administration of a positive allosteric modulator, in a streptozotocin model of T1DM, leads to an improvement of hyperglycaemia and hyperglucagonemia, as well as a reduction of α cell proliferation. Although SIRT-mediated metabolic effects are linked with energy conservation, the author set insulin-dependent pathways to mediate them. In contrast, Codella et al. [86] reported in a non-obese diabetic mouse (NOD) T1DM model a higher α cell ratio after seven weeks of physical training setting in a non-obese diabetic mouse (NOD) T1DM model. This fact highlights the influence of both the onset and severity of diabetes exercise effects on pancreatic architecture. However, anti-islets and anti-glucagon antibodies were the only pancreatic cell markers used, limiting the proliferating pancreatic cell phenotype [86].

Interestingly, Lam et al. [82] found some α cell proliferation in adolescents and young adults with T1DM; however, only one-third had positive glucagon cell markers. The remaining proliferative cells were *Aristaless-related homeobox* (*ARX*) positive (an α cell marker) and sex-determining region Y-box 9 (SOX-9) positive (a development transcription factor). These findings may suggest that α cell mass hyperplasia is not a significant contributing factor to hyperglucagonemia in T1DM. Instead, we may infer a possible plasticity property in β cells, transforming into a non-differentiated form, known as trans-differentiation, or into other pancreatic-endocrine cells, defined as dedifferentiation [87,88].

The basis for understanding these processes begins in pancreatic embryology: After week five, a small group of stem cells from primitive pancreatic duct epithelium differentiate into endocrine cells precursors (HECP) by the expression of Neurogenin 3 (Ngn3) protein [89], a transcription factor encoded by *NEUROG3* gene exclusively expressed in pancreatic HECP. The subsequent expression of other transcription factors defines HECP final phenotype; the FoxO1(+), Pdx1(+), MafA(+), Pax 6(+) and MKx6(+) expression determines β cell differentiation, while Mafβ and Arx1 are associated with α cell phenotype [90]. Trans-differentiation and dedifferentiation processes of β cells have been reported in T1DM and T2DM patients; however, the pathophysiological significance of this finding is still controversial [91,92,93]. *Ex vivo* and *in vivo* studies in human and macaque pancreas have reported that diabetic subjects do not exhibit a reduction in β cells mass, but also an increase in α cell population and α cell/β cells ratio, which may explain why diabetic subjects showed neither reduction in islet size nor hyperglucagonaemic state [91,92,93].

Chronic hyperglycaemia induces glucotoxicity causing islet-chronic inflammation [94,95] and PRC2 deflective function [96], resulting in β cells trans-differentiation via MafA, Nkx6.1, FoxO1, Pdx1 identity markers down-regulation [94,97,98] as well as expression of molecules only found in HEPC like Ngn3, Oct4, L-Myc or aldehyde dehydrogenase-1 isoform A3 (ALDH1A3) [99], coming in the immature, non-secreting form of a pancreatic endocrine cell. However, it is also demonstrated that β cells can be reprogrammed into another type of endocrine cells and even become “bi-hormonal cells” that express insulin and glucagon. Talchai et al. demonstrated the conversion of β cells into α cells in the knock-out FoxO1 mouse model under physiological stress (advanced age and multiple pregnancies) and the simultaneous hyperglycaemia and hyperglucagonemia [94]. In this regard, Spijker et al. [92] demonstrated a significant increase of insulin^+^/glucagon^+^ cells in pancreatic islets and Nkx6.1^+^glucagon^+^insulin−cells in humans and non-human primates with T2DM compared with non-diabetic controls. They established a possible association between high islet amyloidosis and an increased number of Nkx6.1^+^glucagon^+^cells. Gao et al. reported α cell transcriptome expression in β cells, as well as the acquirement of physiological features of α cells following the deletion of the *Pdx1* gene in mice β cells [100]. Interestingly, Malenczyk et al. found that genetic or pharmacological inhibition of secretagogin, a calcium-binding protein involved in the maintenance of β cells identity, are related to its dedifferentiation. Secretagogin in normal conditions prevents *Pdx1*-proteasome degradation [101]. Besides this evidence, it has been shown that glucagon secretion following *Pdx1* deletion is not sustained in time, and therefore, it may not be a significant contributor to hyperglucagonemia [102].

Despite high-quality evidence published recently concerning β cells dedifferentiation into α cells, inconsistent findings have been reported concerning its contribution to systemic hyperglucagonemia [92,93,95,103]. Additionally, limitations related to pancreatic islets’ environment replication in experimental models, and insufficient standardisation in pancreatic cell lineage markers (especially in transitional) states have promoted a sort of scepticism in clinicians. However, future studies will open new avenues that shift our current therapeutic perspective. In this sense, Wang et al. have shown that in vivo studies where glycaemic control is reached by insulin use may revert β cells’ dedifferentiation, obtaining fully functional β cells which secrete insulin in response to sulfonylurea-type drugs but not to hyperglycaemia [95]. Those findings were not reported with sodium/glucose transporter-1 (SGLT-1) inhibitors or thiazolidinediones administration [104,105]. However, not only was the glycaemic control related to decreased β cells dedifferentiation; Ishida et al. reported that regardless of the glycaemic control obtained in diabetic laboratory mice by pair-feeding, insulin, phloridzin, or rosiglitazone administration, only pair-feeding was linked with a reduction of β cells dedifferentiation [105]. These findings reflect that lifestyle changes are the leading therapeutic options in T2DM management, not only for glycaemic control but also for β cells’ mass preservation and possibly against the reduction of α cells mass.

### 4.2. Alpha Pancreatic Cell Dysfunction, Energetic Sensors, and Ionic Channels

Dysfunction in glucose-mediated glucagon releasing mechanisms in α cells is extensively assessed in T2DM [106]. In this regard, Zhang et al. [107], using islets from T2DM organ donors, reported that glucose did not inhibit glucagon secretion. Furthermore, they replicated this methodology in non-diabetic samples by administering diazoxide, a well-known K_ATP_ opener, and oligomycin as an inhibitor of ATP production. Interestingly, in pancreatic islets treated with oligomycin, tolbutamide administration restored glucagon secretion, suggesting that DM hyperglucagonemia is linked to an impaired α cell K_ATP_ inhibition [107].

A recent in vivo study by Knudsen et al. [108] performed on mice lacking fumarase (Fh1) in their β and α cells developed spontaneous T2DM over 10 weeks of age secondary to an impairment of insulin secretion and found a lack of glucagon inhibition on hyper- glycaemic mice. They proposed that hyperglycaemia causes an increment of intracellular sodium in α-cell, secondary to the overactivity of SGLT in these cells. To overcome this, sodium antiporters, as Sodium-H+ transporters, are activated to reduce cellular sodium, conducing to significant cytoplasmic acidification. A subsequent reduction of mitochondrial matrix Ph occurred due to changes in cell electrochemical gradient, carrying an inhibition of oxidative phosphorylation, a decrease of ATP production, a decrease in ATP/ADP ratio, and a consequent opening of K_ATP_, which culminate with an inappropriate glucagon secretion during a hyperglycaemic state. The researchers also hypothesised that mitochondrial and cytoplasmic acidification would lead to a reduction in fumarase activity [108,109], leading to an accumulation of fumarate in the cytoplasm. Thus, glycolysis may contribute to the worsening of the chaotic state of acidification and K_ATP_ activation in α cell during hyperglycaemia. Knudsen et al. [108] also set the inhibition of fumarase and disruption of ATP production may be found in other SGLT-expressing cells. It might be related, in part, to the kidney and cardiac failure observed in diabetes, which is also one of the plausible explanations for the cardiovascular risk and renal benefits of SLGT-1 inhibitors in T2DM patients [110,111].

Impaired Glucokinase (GCK) activity is another metabolic alteration linked to hyperglucagonemia in diabetes. Basco et al. [112] developed an in vivo study on GCK-deficient mice reporting an increase in post-prandial glucagon levels with hyper-glucagonemia, an increment in the liver gluconeogenic gene expression, and therefore, a hepatic glucose output increment. This fact may point to evidence of an underexpression/dysfunction of α-cell-GCK in T2DM, suggesting that GCK enhancers are a promissory therapy, not merely for their effects on insulin but also for their inhibitory effects on glucagon secretion. However, a recent trial with a novel GCK activator, HMS5552, did not report any significant improvement in glucagon levels [113]. Therefore, islet alterations (such as reduced β-cell mass) and α-cell alterations are typical phenomena of long-standing T2DM and possible transformation into undifferentiated cell forms or other islet cells through glucotoxicity. Otherwise, α cell K_ATP_ impairment is hypothesised as a hyperglucagonemia helper and lacks fumarase activity in DM.

## 5. Glucagon as a Therapeutic Target in T2DM

Considering the role of glucagon in diabetes pathogenesis, it could be considered as a promising pharmacological target either by blocking its effect at the receptor level or by the inhibition of its secretion.

### 5.1. Glucagon Receptor Antagonists (GCA): The α Cell as a Therapeutic Target

Hyperglucagonemia effects on glucose metabolism in DM patients are catastrophic, so blocking the systemic effects of glycogen may change the current therapeutic approach in hyperglycaemic syndromes. One of the first developed antagonist molecules were glucagon-derived peptides; however, their effectiveness is limited as they have some partial agonist effects [114]. Thus, other GCA molecules were developed, as non-peptide small molecules, which do not have these undesired effects and are administered by mouth, monoclonal antibodies, or molecules with antisense oligo-nucleotide technology which inhibit the glucagon gene expression by its mRNA blockage. The latter have a better safety profile; however, they are administered subcutaneously (SC) (Table 1). Multiple clinical trials have been developed to search for an ideal GCA. Unfortunately, besides their glycaemic control effectiveness, some of them were discontinued due to significant side effects like LDL-C increase, persistent fatty liver disease, a substantial increase in blood pressure, weight gain, and malignant transformation of α cells [115,116,117]. However, newer GCA drugs have demonstrated an improved safety profile with promising outcomes (Table 1). As glucagon has essential effects on hepatic lipid metabolism (i.e., PPAR-α-dependent pathways), an increase in hepatic lipid content has been linked with GCA-related aspartate aminotransferase (AST) and alanine aminotransferase (ALT) elevation [116,118].

Nevertheless, as non-alcoholic steatohepatitis (NASH) is a common comorbidity in diabetes, it may be considered a possible risk factor for developing this side effect. Another concerning side effect associated is the increase of total cholesterol, and even worse, LCL-C levels. However, recent studies demonstrate that this effect is observed mainly in small molecules’ GCA, increasing cholesterol absorption in the gut [115]. Monoclonal antibodies and antisense oligonucleotides GCA have shown good efficacy with significantly lower side effects (Table 1), but we should wait for the results of phase 3 studies to determine their effectiveness as monotherapy in T2DM. However, one of the concerns regarding GR targeted-therapies is neuroendocrine neoplasia development, as has been described in studies where gene inactivation of these GR in murine models has developed pancreatic neuroendocrine tumours [119]. In the same vein, humans with GR gene inactivating biallelic mutations [120] lead to glucagon cell hyperplasia and neoplasia (Mahvah syndrome) [121].

### 5.2. Incretin Mimetics: Aiming α Cell/β Cells Duality

GLP-1 is an insulinotropic intestinal-derived peptide with glycaemic control properties by enhancing insulin secretion, reduces gastric emptying and glucagon secretion [9]. Multiple studies have demonstrated that GLP-1 analogues administration has beneficial effects over HbA1C levels and glucagon serum levels [122,123,124,125], and their insulinotropic and glucagonostatic effects are considered equal contributors in T2DM glycaemic control [126]. Attractive, those effects are not entirely limited to T2DM. For example, Kuhadiya et al. reported that the addition of 1.2 and 1.8 mg of Liraglutide in T1DM patients over 12 weeks is associated with a reduction of post-prandial plasma glucagon concentration (72 ± 12% and 47 ± 12%, respectively (*p* < 0.05)), significant weight loss, and a decrease in their insulin requirements [127]. Similar findings have also been reported using Pramlintide [128]. In the same vein, Pieber et al. found that the use of liraglutide for four weeks does not overcome the counter-regulatory effects of glucagon during hypoglycaemia events [129].

Multiple randomised, double-blinded studies have also reported positive effects of dipeptidyl peptidase-4 inhibitors (DPP4I) over glucagon serum levels [130], mainly during the post-prandial state [130,131,132,133]. In this sense, Ahrén et al. reported a reduction in prandial glucagon secretion after 2-years of treatment with Vildagliptin [134]. Similar findings have also been found with Sitagliptin addition [135,136,137] to insulin in the T1DM treatment regimen. Interestingly, Awata et al. found that long-term use of Sitagliptin in patients diagnosed with slowly progressive type 1 diabetes (SPIDDM) or latent autoimmune diabetes in adults (LADA) is linked with improved glycaemic control, as well as important preservation of β cells mass. They also suggest that DPP4I drugs would be the ideal choice as initial treatment, especially in the stage of non-insulin-dependency [135].

Finally, dual GLP-1/glucagon agonists have been developed over the past years, showing a significant weight reduction, even bigger than the observed with the isolated use of GLP-1 and a more significant improvement of glycemia and fatty liver disease [138,139,140,141]. Those findings have been related to the significant effects of both molecules on the hypothalamic satiety centre and the reduction of glucagon-hyperglycaemic effects by GLP-1; nevertheless, more research is needed.

Therefore, research into novel drugs acting at GR level represent a potential therapeutic option soon, especially after their cost-benefit is determined, and more information on the potential development of neoplasms is available: However, DPP4i drugs currently represent an accepted option in the management of T2DM, and have even been proposed as candidates in T1DM management.

## 6. Conclusions

Hyperglycaemia has devastating effects on the human body, and its related complications lead to a significant increase in overall mortality and a considerable increment in health system budgets worldwide. As a result, diabetes management strategies have dramatically changed in the past two decades, and, currently, the β-centric focused management is preferred because of the emergency regarding the role of α cell dysregulation in hyperglycaemia development. Hyperglucagonemia in people with diabetes, especially during the post-prandial stage, makes treatment even more challenging. Therefore, an exhaustive comprehension of glucagon physiology and its regulatory pathways is critical in designing drugs that improve the effects of hyperglucagonemia in diabetes. GCA looks like a promising therapeutic option, especially with the new generation drugs in the pipeline, as they have shown a better safety profile.

Moreover, it would be fascinating to see the results from future phase 3 clinical trials and trials showing their effectiveness with other β cells-centred drugs besides metformin. The effects of GLP-1 and DPP4I over glucagon secretion regulation and GLP-1 effects to preserve β cells mass by trans-differentiation and dedifferentiation processes encourage their use in the clinical practice. Glucagon-centred treatments as adjuvant therapy in T1DM have shown promising results, especially in the control of post-prandial hyperglycaemia. These encouraging findings put into perspective the role of glucagon and α cells in diabetes treatment and demonstrate that diabetes is more than a “relative or absolute insulin deficiency”.

## Figures and Tables

**Figure 1 ijms-22-09504-f001:**
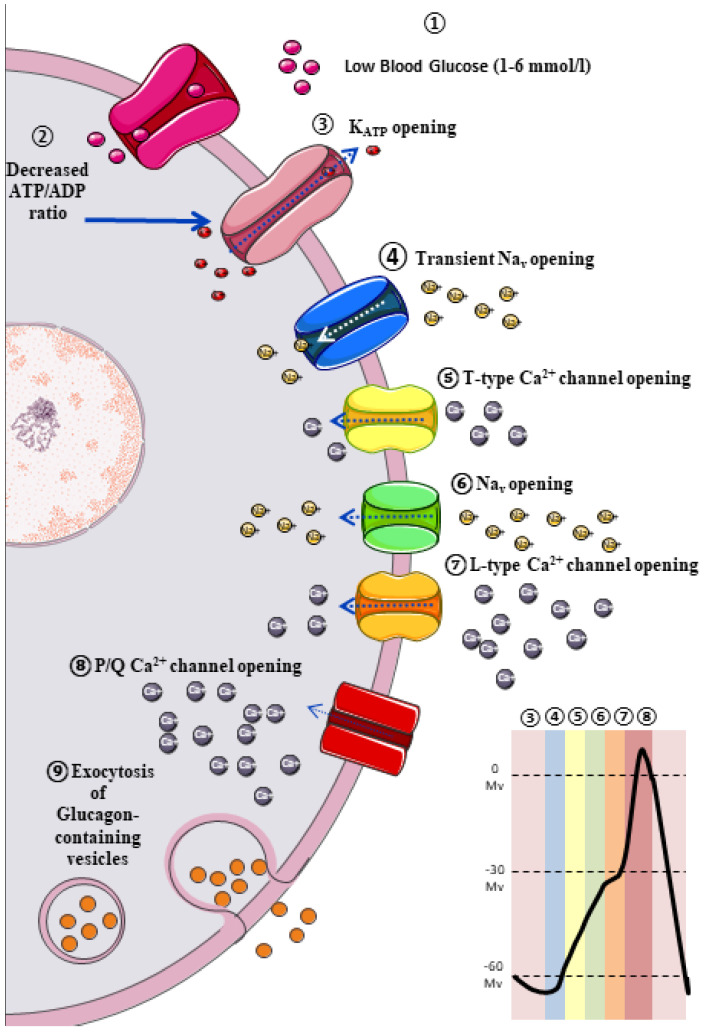
Intrinsic model of glucagon secretion mediated by ATP-sensitive potassium channels (K_ATP_). In conditions of systemic hypoglycaemia (1–6 mmol/L) ATP/ADP ratio in α cells decreases, allowing the opening of K_ATP_ and subsequent reduction of membrane potential, which allow the activation of voltage-gated sodium channels (Na_v_). The increase of intracellular sodium generates a short action potential, given to a rapid inactivation of Na_v_ when membrane potential exceeds −60 mV. This short action potential is enough to allow the opening of T type Ca^2+^ channels, and subsequently the opening of Na_v_, and finally the activation L type Ca^2+^ channels, which generate a positive membrane action potential, allowing the opening of type P/Q Ca^2+^ channels (main Ca^2+^ channel), and thereby generating optimal intracellular calcium levels for glucagon vesicles exocytosis. Finally, this increase in membrane potential would culminate in activation of voltage-gated potassium channels (VGK), and subsequent cellular repolarization.

**Figure 2 ijms-22-09504-f002:**
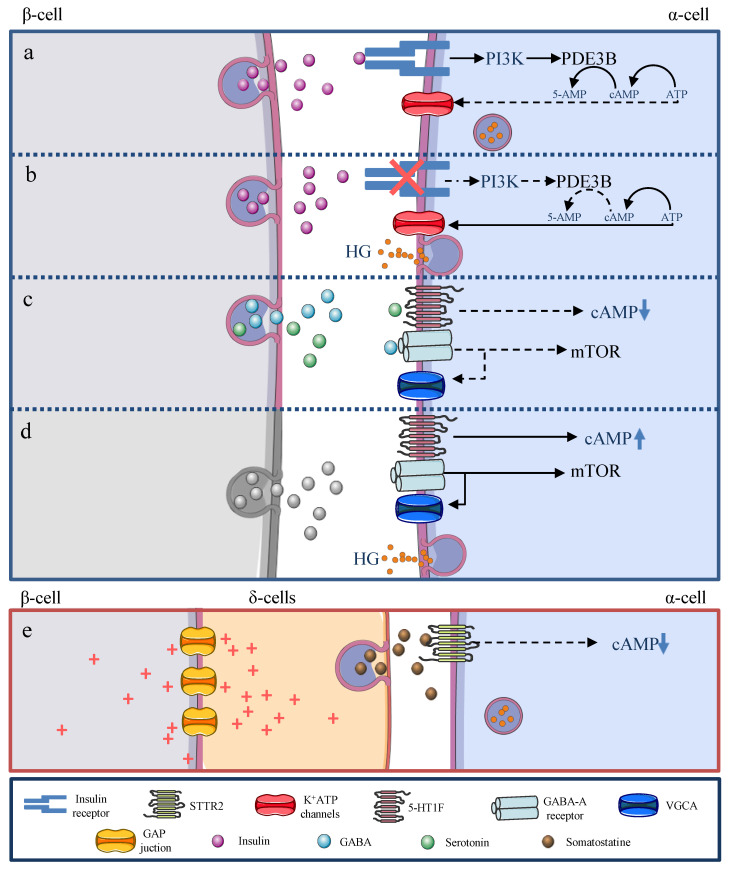
Paracrine regulation of glucagon secretion. (**a**) Insulin signalling in α cell regulates glucagon secretion. It has been proposed that sequential phosphatidylinositol 3-kinase (PI3K) and phosphodiesterase-3B (PDE3B) activation drive a fall in AMP-c concentration and a reduced ATP binding capacity K_ATP_, blocking glucagon secretion. (**b**) The insulin receptor-coding gene deletion in α cell drives to hyperglycaemia and hyperglucagonemia via ATP accumulation. (**c**) GABA and serotonin from β cells inhibit glucagon secretion. GABA-A receptors under non-pathologic conditions trigger α cell hyperpolarization with VGCA closure and rapamycin target kinase in mammalian cells (mTOR) inactivation, one of the primary regulators of α cell proliferation. Serotonin via 5-HT1F membrane receptor activation leads to cAMP reduction, inhibiting, therefore, glucagon release. (**d**) Consequently, any impairment on their secretion could be linked to an increase in α cell mass and hyperglucagonemia in diabetes. (**e**) β cells glucose-mediated depolarization cause coupled- δ-cells depolarization, activating somatostatin secretion, subsequent α cell hyper-polarization, and inhibition of glucagon release by somatostatin receptor 2 (STTR2) activation.

**Figure 3 ijms-22-09504-f003:**
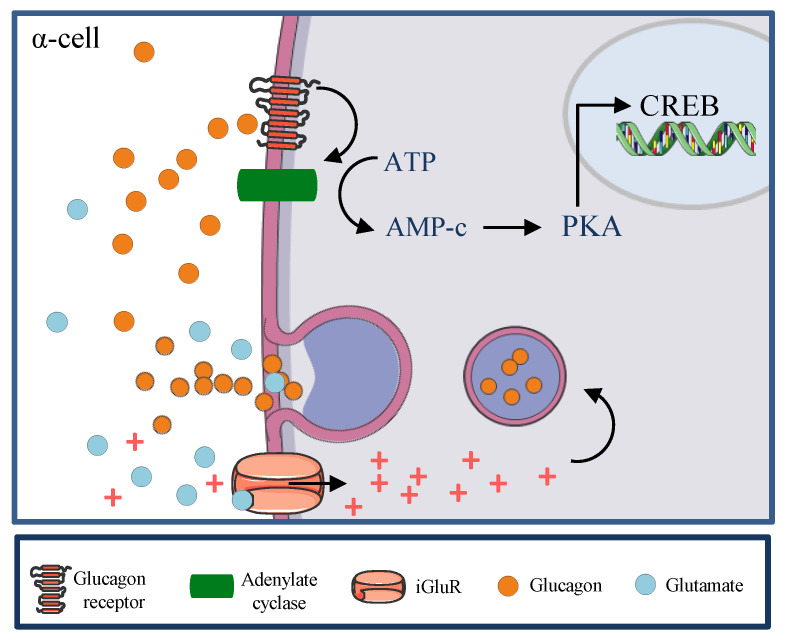
Autocrine regulation of glucagon secretion. In the α cell, GRs activation increases glucagon gene transcription via cAMP response element-binding (CREB). Glutamate is also present in α cell vesicles. It acts as a glucagon-secretion enhancer acting by ionotropic glutamate receptors (iGluR) stimulation or by binding other proteins like α-amino-3-hydroxy-5-methyl-4-isoxazolepropionic acid (AMPA) receptor and kainate receptors in the α cell membrane. Activation of iGluR drives to α cell depolarisation, activation of VGCA, and subsequent the fusion of glucagon-containing vesicles with cell membranes.

**Table 1 ijms-22-09504-t001:** Current glucagon receptor antagonist experimental drugs.

Type	Drug Name	Clinical Trial	Study Design	RA	Results	Main Side Effects	Ref
Small Molecule antagonists	LGD-6972	NCT02250222	Randomised, double-blinded, phase 2 placebo-controlled trial in 166 subjects with T2DM on a stable dose of metformin for 12 weeks.	Oral	MB HbA1C 8.2%, HbA1c was reduced −0.90% (5 mg), −0.92% (10 mg), −1.20% (15 mg), −0.15 (placebo)	Mild increase AST/ALT Non-severe hypoglycaemia	[118]
	PF-06291874	NCT02175121	Randomised, double-blind, stratified, placebo-controlled, 4-arm, parallel-group study × 12 weeks in 206 adults with T2DM on stable doses of metformin	Oral	MB HbA1C: 8.2%, HbA1c was reduced −0.67% (30 mg), −0.91% (60 mg), −0.93% (100 mg), and increased 0.25% in placebo group	Blood pressure elevation. Mild increase AST/ALT Mild LDL-C increase weight gain	[122]
Monoclonal antibody antagonist	RN909	NCT02211261	Phase 1, placebo-controlled, randomised, single- and multiple-ascending dose study in 84 T2DM subjects receiving stable metformin regimens every four weeks for 12 weeks.	SC	HbA1c was reduced a mean of −0.83% to −1.56% with RN909 versus 0.10% with placebo in all single-dose and multiple-dose RN909 treatment groups	Mild increase AST/ALT Blood pressure elevation	[123]
	REMD-477	NCT02715193	Phase 1, randomised, placebo-controlled, double-blinded, performed in 21 patients with T1DM. They received a REMD-477 single dose of 70 mg and eight weeks follow-up.	SC	Decreased by 14% insulin requirements in case group during post-treatment days 6–12. 27 mg/dl average reduction of daily glucose concentration	Headaches Oropharyngeal pain	[124]
Antisense oligonucleotide targeting the GR	ISIS-GCRRx (Also known as IONIS-GCGRRx)	NCT01885260 NCT02583919 NCT02824003	In three phase-2, randomised, double-blind studies, patients with T2DM on metformin received weekly SC injections of IONIS-GCGRRx (50–200 mg) or placebo for 13 or 26 weeks.	SC	MB 8.6–8.9%.%. HbA1c was reduced at week 14 −2.0% (200 mg), −1.4% (100 mg), −0.3% (placebo), and −1.6% (75 mg), −0.9% (50 mg), −0.2% placebo at week 27 A significant dose-dependent increase of GLP-1 also was found.	Mild increase AST/ALT Increase in hepatic lipid content Local reactions at the injection site	[125]

RA: route of administration; T2DM: type 2 diabetes mellitus; T1DM: type 1 diabetes mellitus; MB: mean baseline; HbA1C: glycosylated haemoglobin; AST: aspartate aminotransferase; ALT: alanine aminotransferase; LCL-C: low-density lipoprotein. GLP-1: glucagon-like peptide-1; SC: subcutaneous.

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
