# Peer review of "The Role of the α Cell in the Pathogenesis of Diabetes: A World beyond the Mirror"

_ijms, 2021, doi:10.3390/ijms22179504_

Round 1
Reviewer 1 Report
Dear Authors,
Your thorough and comprehensive review of the role of glucagon in the glucose metabolism is interesting and is surely worth to be published. I found only some minor issues:
- abbreviations should be explained at their first occurence
- expansion of the abbreviation SGLT-1 is sodium/glucose transporter-1
- some small language errors should be corrected.
I have no major remarks.
Best Regards

Author Response
"Please see the attachment."

Reviewer 2 Report
In this manuscript Martínez et al review the role of a cells in the pathogenesis of diabetes, while discussing the mechanisms that may be involved in glucagon hypersecretion. The authors discuss how glucagon-induced insulin resistance aggravates the metabolic consequences of the insulin-deficient state and promotes diabetes. The authors slightly referred to the hepatic consequences and pointed some promising therapeutic options that take in consideration the regulation of glucagon secretion.
The scope of the manuscript is very pertinent since glucagon induces insulin resistance that worsens the insulin-deficient condition of T1D and has a pathogenic role in T2D. The comprehension of glucagon metabolic pathways and their regulation is critical for the design of effective drugs to manage the effects of hyperglucagonemia in diabetes.
This manuscript could be made into a very good much shorter paper.
The manuscript should be thoroughly revised and edited to correct typos and poor sentences.
The introduction section should detail the rationale for the organization of the revision, giving a brief explanation, contextualization and justification regarding the choice of the main sections that compose the whole manuscript.
At the end of each section should be presented the main point to be extract/ main conclusion.
The text should be standardized. Please always use the same way of referring, for example, to alpha cells, beta cells, ATP sensitive potassium channels, calcium, type 1 diabetes, type 2 diabetes etc. The same abbreviation must be used in all text sections, abstract, figures, figure legends and in the table.
Example of unequal designation
Line 36 K+/ATP channel opening
Line 181 ATP-sensitive potassium channels (K+ATP-channels)
Figure 1 KATPC
Line 204 Figure legend ATP-dependent potassium channels (KATPC)
Suggestion for the first mention
Alpha (a) cell, become “a cell” and is keep constant throughout the document
Beta (b) cell
Delta (D) cell
Epsilon (e) cell
Type 1 diabetes mellitus (T1D or T1DM)
Type 2 diabetes mellitus (T2D or T2DM) instead to “DM2”
ATP-sensitive potassium channels (KATP or KATP)
voltage-gated sodium channels (Nav or Nav)
Ca++, replace by Ca2+ (upperline)
K+, replace by K+ (+ in upperline)
Glucagon, glucagon no capitals when not at the beginning of the sentence
Insulin, insulin, no capitals when not at the beginning of the sentenc
In addition, there are inaccuracies and vague claims that must be detailed and resolved, some examples :
Line 50 “From a general perspective, the canonical pathophysiological DM2 events state that insulin resistance develops when a progressive ectopic fat deposition occurs in both liver and muscle occurs because of dysfunctional visceral adipose tissue due to chronic hypoxia, inflammation, and oxidative stress” please make clear
Line 87 “diabetes, given a valuable step to realise that realised the”
Line 93 “A cells”… “beta cells” please keep consistent the abbreviations use
Line 99 “University of Toronto Ref “ Typo?
Line 121 “Decade after, in 1975, Orcid and Unger developed the classical cellular arrangement 121 model of pancreatic islets” please improve the sentence
Line 126 “b-cells”
Line 127 “a-cells”
Line 133 “The human pancreas contains 1–2 million islets, each of 50−100μm in diameter, and constituted by ∼2000 cells on average. Although this number seems to be exceptionally large, these cells are only the 2% of the overall pancreatic mass. It is worthy to note that 65 % of the human islets cells are α-cells [30]”. Please be consistent in spacing between number and units
Line 143 “The distribution of alfa, beta, delta, PP, and epsilon cells within islets is …”
Line 145 Please improve the sentence “This fact has significant functional consequences for the differences in paracrine regulation between human and rodent islets. Therefore, any alteration on this cellular network may dysregulate their secretory patterns and as a result abnormal glucose metabolism.”
Line 168 – Please improve the sentence “The cellular machinery controlling the secretion of glucagon in α-cells is remarkably 168 like that regulating the secretion of Insulin in β-cells,”
Line 170 – Please improve the sentence “Langerhans's islets are well vascularised to ensure rapid glucose sensing in line with nutrients like glucose and amino acids are the major determinants of Glucagon (and insulin) secretion.”
Line 178 “acetyl-CoA oxidation in the Krebs' cycle and finally, adenosine triphosphate (ATP) synthesis by oxidative phosphorylation in the electronic transport chain throughout the thickness of the inner mitochondrial membrane”, This statement is scientifically wrong. ATP synthesis occurs in ATP synthase, that does not belong to the ELECTRON transport chain. Please correct
Figure 1 use in the construction of the figure the same abbreviations that are used in the text
Line 206 “voltage-gated sodium channels (VGNA)… Line 209 “opening of Nav channels (of Na+)”
Line 214 “mecha-nism”
Line 258 “between β-cells and d-cells. Β-cells ...”
Line 262 “two-pore domain K+(K2P) channel TALK-1”
Line 281 improve sentence “It is well-known that α-cell membranes are plenty of glucagon receptors [58], [66].”
Line 292 “and facilitating the glucagon pass to the systemic circulation” …the glucagon passage…
Line 295 Start with capital letter after period
Line 306 Improve the sentence “Centered on discovering a roughly 50% drop in ephrin-A5 immunofluorescence in human islets obtained from patients with DM2, it has been hypothesized that all this mechanism might be included in the hyperglucagonemia seen in diabetes [75].
Line 323 Improve the sentence “Other types of juxtacrine interaction have been suggested to regulate glucagon se- 323 cretion, but they have received less research.”
Line 335 “IV glucose…” – first use, please add the full description
Line 337 glucagon-like peptide-1 (GLP-1) was first mentioned in Line 154
Line 340 Improve the sentence “However, their activity on α-cells is still controversial, given the complexity of their studies on the experimental level [83, 84]”.
Line 348 “change inhibiting P/Q-type Ca2+channels, and therefore blockade of glucagon exocytosis”
Line 362 Please improve the sentence “[91] becoming in DM2”
Line 371 “secre-tion”
Line 381 please format the references “[96]–[98]” to become [96–98], the same for the rest of the document
Line 389 “alfa-cell ratio…”
Line 390 “se-verity”
Line 403/404 “em-bry-ology”
Line 405 “epi-thelium”
Line 409 “beta-pancreatic”
Line 429 “com-pared”
Line 442 “in-sufficient”
Line 465 “K+ATPCh.”
Line 466. Improve the sentence “A recent in vivo study by Knudsen et al. [121] performed on mice lacking fumarase (Fh1) in their β and α cells, developed spontaneous DM2 over ten weeks of age secondary to an impairment of insulin secretion and found a lack of glucagon inhibition on hyper- glycaemic mice.”
Line 474 “pro-duction”
Line 486 “hy-perflucagonemia”
Line 490 “un-der-expression…”
Line 496 “con-sidered”
Line 499 “cata-strophic”
Line 500 “thera-peutic”
Line 503 “devel-oped”
Line 504 missing space after molecules “small molecules,(which do”
Line 509 “discon-tinued”
Line 511 format references
Line 511 Please improve the sentence “However, newer GCA prototypes have shown a better safety profile with promising results soon”
Line 515 “ami-notransferase”
Line 523 “effect-tiveness”
Line 524 Table, placing horizontally would make reading easier, please correct typos. It is missing information in the legend, ex T1D.
Line 529 “se-cretion”
Line 536 “as-sociated”
Line 547 “Sitagliptin [146-148]” citation done properly, please changes throughout the document
Line 544 unquote
Line 554 “devel-oped”
Line 556 Improve sentence “ The strong effects of both molecules on the hypothalamic satiety center and the mitigation of Glucagon-hyperglycaemic effects by GLP-1 have been linked with those findings; however, they are still under study.”
Please add references to the sentences, lines 254, 337, 374, 393, 412 and, 465
Please discuss some of the side effects of the therapies for glucagon control. For example, monoclonal antibodies that block glucagon receptor are beneficial to control diabetes in patients with T1D and T2D. However, a theoretical concern that might be associated to glucagon receptor blockade is development of pancreatic neoplasia, as patients with biallelic inactivating mutations in the glucagon receptor gene (GCGR) develop α-cell hyperplasia that may progress to neoplasia (Mahvash syndrome).

Author Response
"Please see the attachment."
